# MMPareto: Innocent Uni-modal Assistance for Enhanced Multi-modal Learning

## Abstract

Multi-modal learning methods with targeted uni-modal constraints have exhibited their superior efficacy in alleviating the imbalanced multi-modal learning problem, where most models cannot jointly utilize all modalities well, limiting their performance. However, in this paper, we first identify that there are gradient conflict between multi-modal and uni-modal learning objectives, potentially misleading the optimization of shared uni-modal encoders. The necessity of diminishing conflict during gradient integration naturally accords with the idea of Pareto methods, which could provide a gradient that benefits all objectives. Unfortunately, conventional Pareto method surprisingly fails in the context of multi-modal scenarios. We further theoretically analyze this counterintuitive phenomenon and attribute it to the priority of Pareto method for multi-modal gradient with small magnitude, weakening model generalization. To this end, we propose MMPareto algorithm, which could ensure a direction that is common to all learning objectives while preserving magnitude with guarantees for generalization, providing innocent uni-modal assistance for primary multi-modal learning. Finally, empirical results across multiple dataset with different modalities indicate our superior method performance. The proposed method is also expected to facilitate multi-task cases with a clear discrepancy in task difficulty, demonstrating its scalability.

## 1 Introduction

People are immersed in a variety of sensory messages, encompassing sight, sound, and touch, which has sparked the study of multi-modal learning (Baltrušaitis et al., 2018). Although these methods have revealed effectiveness, recent studies have found the imbalanced multi-modal learning problem, where most multi-modal models cannot jointly utilize all modalities well, limiting their performance (Huang et al., 2022). Under this scenario, several methods have been proposed to improve the training of worse learnt modality with additional module or modality-specific training strategy (Peng et al., 2022; Wu et al., 2022). These methods often have one common sense that targetedly improves uni-modal training. Among them, multitask-like methods that directly add uni-modal constraints besides multi-modal joint learning objective, exhibit their superior effectiveness for alleviating this imbalanced multi-modal learning problem (Wang et al., 2020; Du et al., 2023; Fan et al., 2023).

However, behind the effective performance, we observe a hidden risk in model optimization under this multitask-like scenario, potentially limiting the model ability. Every coin has two sides. Uni-modal constraints undeniably enhance the learning of corresponding modalities, alleviating the imbalanced multi-modal learning problem. Meanwhile, the optimization of parameters in uni-modal encoder is influenced by both multi-modal joint learning objective and its own uni-modal learning objective. This entails the need to minimize two learning objectives concurrently, but usually, there does not exist a set of parameters that could satisfy this goal. Consequently, these multi-modal and uni-modal learning objectives could have conflict during optimization. In Figure 1a, we take an example of the video encoder on Kinetics Sounds dataset. Based on the empirical results, negative cosine similarity indicates that multi-modal and uni-modal gradients indeed have conflicts in direction during the optimization of shared uni-modal encoder. Especially, these conflicts at the early training stage could substantially harm the model ability (Liu et al., 2020). Therefore, the primary multi-modal learning is potentially disturbed. Addressing such conflicts is an essential problem that needs to be solved.

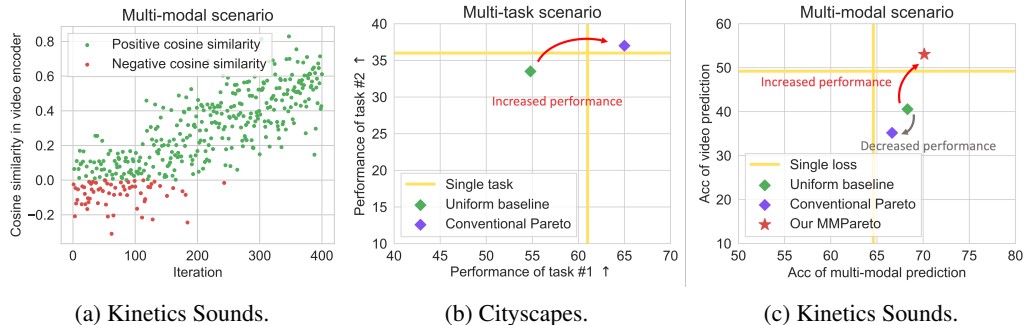

Figure 1: **(a).** Cosine similarity between multi-modal and uni-modal gradients in the video encoder of Kinetics Sounds dataset (Arandjelovic & Zisserman, 2017). **(b).** Methods performance on the multi-task dataset, Cityscapes (Cordts et al., 2016). Results are from Sener & Koltun (2018) **(c).** Methods performance of multi-modal and uni-modal prediction on Kinetics Sounds. Single task/loss is the results of solely trained model with one learning objective.

To avoid optimization conflicts, it is essential to well integrate both gradients, making the uni-modal gradient not affect the primary multi-modal training but assist it. This necessity naturally accords with the idea of Pareto method (Sener & Koltun, 2018), which aims to find a steep gradient that benefits all objectives and finally converges to a trade-off state of them. This trade-off is called Pareto optimal (Pareto, 1897), in which no objective can be advanced without harming any other objectives. As shown in Figure 1b, in the typical multi-task field, Pareto method has achieved ideal advancement in balancing the learning objective of multiple tasks. Therefore, it is expected to keep superiority in solving conflicts in this multitask-like multi-modal learning framework. However, the fact is contrary to the assumption. Based on the results in Figure 1c, the conventional Pareto method surprisingly loses its efficacy, even is clearly worse than the uniform baseline without any gradient integration strategy, where all gradients are equally summed.

To delve into this counterintuitive phenomenon, we first analyze the property within multi-modal learning scenarios and the conventional Pareto method. Concretely, different from typical multi-task cases, optimization of multi-modal joint loss with information from all modalities is often easier than uni-modal loss only with information from one modality. Accordingly, the optimization of multi-modal loss is favored over uni-modal loss, since the greedy nature of deep neural network (Mandt et al., 2017). Therefore, multi-modal learning objective with a smaller loss value brings a smaller gradient magnitude than that of uni-modal one. By coincidence, Pareto method tends to pay more attention to the gradient with a small magnitude, resulting in the final gradient after integration being mostly influenced by multi-modal gradient. Therefore, although avoiding conflicts, the efficacy of uni-modal assistance is diminished as well. To go a step further, we theoretically explore the potential influence for model generalization. Our finding suggests that this priority for multi-modal gradient with small magnitude could basically weaken model generalization ability.

Based on the above theoretical analysis, it becomes imperative to address gradient conflicts in the context of multi-modal scenarios. To this end, we propose the **M**ulti-**M**odal **Pareto** (MMPareto) algorithm, which respectively takes into account the *direction* and *magnitude* during gradient integration. It ensures innocent uni-modal assistance, where the final gradient is with direction common to all learning objectives while preserving magnitude with guarantees for generalization. We provide theoretical evidence of the method's desirable convergence properties, demonstrating its ability to reach a Pareto stationarity. Overall, our method diminishes the potential conflict with guaranteed generalization, effectively alleviating the imbalance multi-modal learning problem. As shown in Figure 1c, our MMPareto method provides both advanced multi-modal performance and uni-modal performance. What's more, the uni-modal performance is even superior to the solely trained uni-modal model.

In a nutshell, our contribution is three-fold. **Firstly,** we observe the potential gradient conflict in the effective multitask-like framework for the imbalanced multi-modal learning problem. **Secondly,** we theoretically analyze the failure of Pareto integration in the context of multi-modal learning, and then propose the MMPareto algorithm which could provide innocent uni-modal assistance whose gradients are with non-conflict direction and generalization guaranteed magnitude. **Thirdly**, experiments across different dataset empirically verify our theoretical analysis as well as superior algorithm performance. Furthermore, the proposed method could extend to multi-task cases with clear discrepancy in task difficulty, indicating its scalability.

## 2 RELATED WORK

### 2.1 IMBALANCED MULTI-MODAL LEARNING

Recent research has uncovered the imbalanced multi-modal learning problem, as multi-modal models tend to favor specific modalities, thereby constraining their overall performance (Peng et al., 2022; Huang et al., 2022). Several methods have been proposed for this problem, with a shared focus on targetedly enhancing optimization of each modality (Wang et al., 2020; Peng et al., 2022; Wu et al., 2022; Fan et al., 2023). Among them, multitask-like methods that directly incorporate targeted uni-modal constraints have demonstrated superior effectiveness (Wang et al., 2020; Du et al., 2023; Fan et al., 2023). However, under this multitask-like framework, optimization of uni-modal encoder is simultaneously controlled by the multi-modal joint learning objective and corresponding uni-modal learning objective, which could cause gradient conflict, potentially harming the primary multi-modal learning. In this paper, we observe and diminish the potential conflict by the proposed MMPareto algorithm. Our method could effectively alleviate the imbalanced multi-modal learning problem, achieving considerable improvement.

### 2.2 PARETO INTEGRATION IN MULTI-TASK LEARNING

Shared parameter representation in multi-task learning is expected to fit several learning objectives at once, but usually, there does not exist a single solution that minimizes all objective functions simultaneously, resulting in the potential conflict problem during optimization. To solve conflict, the Pareto method is introduced to integrate different gradients, which aims to find a gradient common to all objectives and finally converge to a trade-off state of them (Sener & Koltun, 2018). Besides the conventional one, the idea of Pareto integration is extended from different perspectives, including more different trade-offs among different tasks or faster convergence speed, to better benefit multi-task learning (Lin et al., 2019; Ma et al., 2020; Liu et al., 2021). Similarly, in this paper, we observe the optimization conflict in the shared uni-modal encoder in the multitask-like multi-modal framework. Inspired by the former success of Pareto integration, we introduce the idea of Pareto integration into multi-modal learning but surprisingly find it failed. To this end, we further theoretically analyze and find the harmed generalization of Pareto integration in the context of multi-modal learning, and then propose MMPareto algorithm which could handle multi-modal scenarios as well as multi-task cases with clear discrepancy in task difficulty.

## 3 METHOD

### 3.1 MULTITASK-LIKE MULTI-MODAL FRAMEWORK

In multi-modal learning, models are expected to produce correct predictions by integrating information from multiple modalities. Therefore, there are often **multi-modal joint loss** in multi-modal framework, which takes prediction of fused multi-modal feature. However, based on recent studies, only utilizing such joint loss to optimize all modalities together could result in the optimization process being dominated by one modality, leaving others being severely under-optimized (Peng et al., 2022; Huang et al., 2022). To overcome this imbalanced multi-modal learning problem, introducing **uni-modal loss** which targets the optimization of each modality is natural. Then, the final learning objective of these multitask-like multi-modal models is:

$$\mathcal{L} = \mathcal{L}_m + \sum_{k=1}^{n} \mathcal{L}_u^k, \tag{1}$$

where $\mathcal{L}_m$ is the multi-modal joint loss and $\mathcal{L}_u^k$ is the uni-modal loss for modality $k$. $n$ is the number of modalities. We mainly consider the multi-modal discriminative task, and both multi-modal joint loss and uni-modal loss are cross-entropy loss functions. The framework is illustrated in Figure 2.

Uni-modal losses in this multitask-like multi-modal framework effectively improve the learning of corresponding modalities, getting rid of the suppression of dominant modality and further alleviating the imbalanced multi-modal learning problem. However, the optimization of parameters in uni-modal encoder is guided by both multi-modal joint learning objective and its own uni-modal learning objective. Concretely, when using *Stochastic Gradient Descent* (SGD) optimization, at iteration

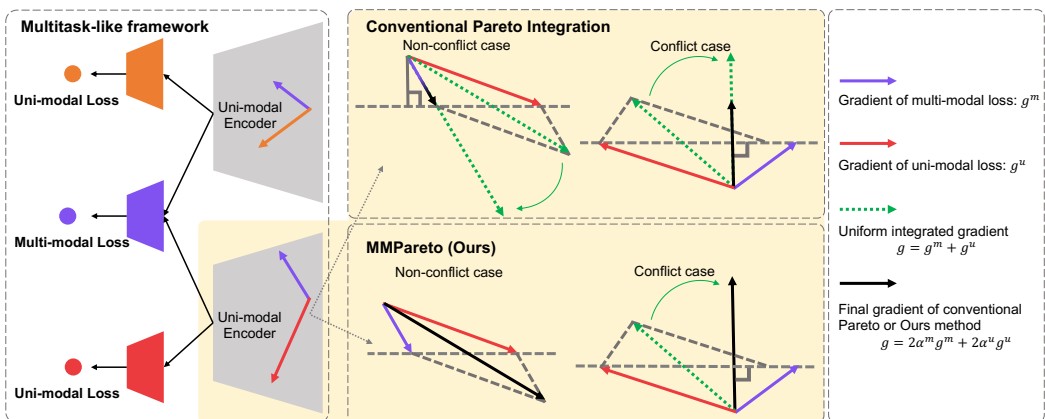

Figure 2: **Illustration of gradient integration strategy of conventional Pareto and our MMPareto.** Conventional Pareto integration has prior for the multi-modal gradient with smaller magnitude. Our MMPareto method avoids gradient conflict and is with generalization guaranteed magnitude.

$t$ with mini-batch $S$, the update of $k$-th uni-modal encoder parameter $\theta^k$ is determined by both $\mathbf{g}_S^m(\theta^k(t))$ and $\mathbf{g}_S^u(\theta^k(t))$, which are the multi-modal gradient and uni-modal gradient, respectively. Each gradient provides the ideal direction for decreasing the corresponding loss function. For brevity, we rewrite $\{\mathbf{g}_S^i(\theta^k(t))\}_{i \in \{m,u\}}$ with $\{\mathbf{g}^i\}_{i \in \{m,u\}}$ in the rest parts.

In multi-modal scenarios, multi-modal loss and uni-modal loss are tightly related, but their gradients may still exist conflicts, as shown in Figure 1a. Here we take an example of the video encoder on Kinetics Sounds dataset. Negative cosine similarity demonstrates that these two learning objective indeed has conflict in direction, especially at the early training stage. Based on the former studies (Liu et al., 2020), disturbances at the early stage could substantially harm the model ability (Liu et al., 2020). Therefore, how to resolve these conflicts and well integrate $\mathbf{g}^m$ and $\mathbf{g}^u$ needs to be solved.

## 3.2 PARETO INTEGRATION IN THE CONTEXT OF MULTI-MODAL LEARNING

### 3.2.1 PARETO INTEGRATION

To avoid these optimization conflicts within uni-modal encoders disturbing the primary multi-modal learning, it is essential to well integrate different gradients without bringing negative effects on multi-modal training. This goal accords with the idea of Pareto method in multi-task learning (Sener & Koltun, 2018). In Pareto method, at each iteration, gradients are assigned different weights, and the weighted combination is the final gradient, which can benefit all learning objectives. Finally, parameters can converge to a trade-off state, Pareto-optimality, in which no objective can be advanced without harming any other objectives. It is natural to introduce Pareto integration into multi-modal framework, avoiding conflict between multi-modal and uni-modal gradients. Concretely, in our case, the Pareto algorithm is formulated to solve:

$$\min_{\alpha^m, \alpha^u \in \mathcal{R}} \|\alpha^m \mathbf{g}^m + \alpha^u \mathbf{g}^u\|^2$$
$$s.t. \quad \alpha^m, \alpha^u \geq 0, \alpha^m + \alpha^u = 1, \tag{2}$$

where $\|\cdot\|$ denotes the $L_2$-norm. This problem is equal to finding the minimum-norm in the convex hull of the family of gradient vectors $\{\mathbf{g}^i\}_{i \in \{m,u\}}$. We denote the found minimum-norm as $\omega$. Désidéri (2012) showed that either $\omega$ to this optimization problem is $0$ and the corresponding parameters are Pareto-stationary which is a necessary condition for Pareto-optimality, or $\omega$ can provide descent direction common to all learning objectives.

### 3.2.2 GENERALIZATION HARMED RISK OF PARETO

Pareto integration is expected to exhibit advantage under multitask-like multi-modal framework, since gradient conflicts are resolved. However, based on Figure 1c, the conventional Pareto method surprisingly fails, and is clearly worse than the uniform baseline case without any specifically designed gradient integration strategy. To explore the hidden reason, we further analyze the property within multi-modal learning and the Pareto integration method.

Although with a similar framework, our multi-modal case is different from the typical multi-task ones. Uni-modal loss only receives the prediction based on data of corresponding modality. In contrast, multi-modal loss with prediction based on data of all modalities is easy to be optimized with more adequate information. When simultaneously optimizing both learning objectives, the multi-modal loss is favored over uni-modal one, since the greedy nature of deep neural network (Mandt et al., 2017). Hence multi-modal loss with a smaller loss value leads to a smaller gradient magnitude than that of uni-modal one. In other words, $\|\mathbf{g}^m\| < \|\mathbf{g}^u\|$. Combined with this discrepancy in gradient magnitude, for the optimization problem of Equation 2, we can have its analytic solution:

$$
\begin{cases}
\alpha^m = 1, \alpha^u = 0 & \text{if } \cos\beta \geq \frac{\|\mathbf{g}^m\|}{\|\mathbf{g}^u\|}, \\
\alpha^m = \frac{(\mathbf{g}^u - \mathbf{g}^m)^\top \mathbf{g}^u}{\|\mathbf{g}^m - \mathbf{g}^u\|^2}, \alpha^u = 1 - \alpha^m & \text{else.}
\end{cases}
$$

$\beta$ is the angle between $\mathbf{g}^m$ and $\mathbf{g}^u$. During training of the current iteration, the gradient weight of $\mathbf{g}^m$ and $\mathbf{g}^u$ are assigned as $2\alpha^m$ and $2\alpha^u$, respectively[1]. Here we further analyze the Pareto analytic solution. When $\cos\beta \geq \frac{\|\mathbf{g}^m\|}{\|\mathbf{g}^u\|}$, we have $\alpha^m > \alpha^u$. Otherwise, since $\|\mathbf{g}^m\| < \|\mathbf{g}^u\|$, we have:

$$
\begin{aligned}
\alpha^m - \alpha^u &= \frac{(\mathbf{g}^u - \mathbf{g}^m)^\top \mathbf{g}^u}{\|\mathbf{g}^m - \mathbf{g}^u\|^2} - \left(1 - \frac{(\mathbf{g}^u - \mathbf{g}^m)^\top \mathbf{g}^u}{\|\mathbf{g}^m - \mathbf{g}^u\|^2}\right) \\
&= \frac{\|\mathbf{g}^u\|^2 - \|\mathbf{g}^u\|\|\mathbf{g}^m\|\cos\beta}{\|\mathbf{g}^m - \mathbf{g}^u\|^2} - \frac{\|\mathbf{g}^m\|^2 - \|\mathbf{g}^u\|\|\mathbf{g}^m\|\cos\beta}{\|\mathbf{g}^m - \mathbf{g}^u\|^2} \\
&> 0. \quad (\|\mathbf{g}^m\| < \|\mathbf{g}^u\|)
\end{aligned}
\tag{3}
$$

**Remark 1.** *Conventional Pareto integration method tends to assign larger weight to the gradient with smaller magnitude.*

As stated in Remark 1, we can conclude that the Pareto method tends to assign larger weight to the multi-modal gradient $\mathbf{g}^m$ during integration.

To further explore influence of this priority, we first analyze and obtain the SGD generalization bound for case without any gradient integration strategy. Denote the expected risk as $\mathcal{R}(Q)$ and empirical risk as $\hat{\mathcal{R}}(Q)$. Concretely, for any positive real $\sigma \in (0, 1)$, with probability at least $1 - \sigma$ over a training sample size of size $N$, we have:

$$
\mathcal{R}(Q) \leq \hat{\mathcal{R}}(Q) + \sqrt{\frac{\frac{\eta}{|S|}\text{tr}\left((C^m + C^u)A^{-1}\right) - 2d - 2\log(\det(\Sigma)) + 4\log\left(\frac{1}{\delta}\right) + 10\log N + 32}{8N - 4}},
\tag{4}
$$

where $\eta$ is the learning rate and $S$ is the set of mini-batch. $C^m$ and $C^u$ are the covariance matrix, bringing by the random sampling. $A$ is the Hessian matrix around the optimum.

Furthermore, when applying Pareto integration, based on Remark 1, the Pareto method tends to assign larger weight for gradient with smaller magnitude. Accordingly, the final gradients of Pareto method would always have a smaller magnitude than the uniform baseline without any gradient integration strategy. Then, we further obtain the SGD generalization bound with Pareto gradient integration. For any positive real $\sigma \in (0, 1)$, with probability at least $1 - \sigma$ over a training sample size of size $N$, we have:

$$
\mathcal{R}(Q) \leq \hat{\mathcal{R}}(Q) + \sqrt{\frac{\frac{\eta\gamma}{|S|}\text{tr}\left((C^m + C^u)A^{-1}\right) - 2d - 2\log(\det(\Sigma)) + 4\log\left(\frac{1}{\delta}\right) + 10\log N + 32}{8N - 4}},
\tag{5}
$$

where $0 < \gamma < 1$ is the least magnitude difference between Pareto integration and the uniform baseline among all training iterations. Based on these two generalization bounds, the model generalization ability is weakened after applying Pareto integration, compared with the case where all gradients are not specifically adjusted, since $0 < \gamma < 1$. Moreover, here $\gamma$ is the least magnitude difference between Pareto integration and the case without specific gradient integration. Therefore, Equation 5 is in fact a loose bound. Model generalization would be affected more in practice. The detailed proof is provided in Appendix C. This theoretical analysis about the harmed generalization of conventional Pareto method explains the results in Figure 1c, where it surprisingly loses its efficacy, even is clearly worse than the uniform baseline without any gradient integration.

---

[1]Here we use $2\alpha^i$ as the gradient weight is to keeps the same gradient weight summation with uniform baseline without any specifically designed gradient integration strategy, where all weight is assigned as 1.

### 3.2.3 Multi-modal Pareto algorithm

Based on the above theoretical analysis, conventional Pareto method has priority for gradient with smaller magnitude. Hence its final gradients always have a smaller magnitude than that of the case without any gradient integration strategy. Accordingly, the model generalization ability is basically weakened. Therefore, directly introducing Pareto integration in the context of multi-modal learning is unreasonable and invalid. It becomes essential to address gradient conflicts in multi-modal scenarios.

Hence we propose the **Multi-Modal Pareto** (MMPareto) algorithm. It considers both the *direction* and *magnitude* during gradient integration, to provide innocent uni-modal assistance where the final gradient is with direction common to all learning objectives while preserving magnitude with guarantees for generalization. Our method considers the conflict case and non-conflict case respectively. The overall algorithm is shown in Algorithm 1 and illustrated in Figure 2.

**Non-conflict case.** We first consider the case $\cos\beta \geq 0$. Under this case, the cosine similarity between $\mathbf{g}^m$ and $\mathbf{g}^u$ is positive. For the direction, the arbitrary convex combination of the family of gradient vectors $\{\mathbf{g}^i\}_{i\in\{m,u\}}$ is common to all learning objectives. Therefore, to maintain the gradient magnitude as the case without specific gradient integration, we assign $\alpha^m = \alpha^u = \frac{1}{2}$. With this setting, the risk of weakening model generalization is avoided.

**Conflict case.** For the case $\cos\beta < 0$, it is essential to find the direction that

---

**Algorithm 1** MMPareto

**Require:** Training dataset $\mathcal{D}$, iteration number $T$, initialized uni-modal encoder parameters $\theta^k$, $k \in \{1, 2, \cdots, n\}$, other parameters $\theta^{\text{other}}$.
  **for** $t = 0, \cdots, T-1$ **do**
    Sample a fresh mini-batch $S$ from $\mathcal{D}$;
    Feed-forward the batched data $S$ to the model;
    Calculate gradient using back-propagation;
    Update $\theta^{\text{other}}$ without gradient integration method;
    **for** $k = 1, \cdots, n$ **do**
      Obtain $\mathbf{g}^m$ and $\mathbf{g}^u$ for $k$-th uni-modal encoder;
      Calculate $\cos\beta$; $\beta$ is angle between $\mathbf{g}^m$ and $\mathbf{g}^u$;
      Solve problem of Equation 2, obtain $\alpha^m$, $\alpha^u$;
      **if** $\|\alpha^m\mathbf{g}^m + \alpha^u\mathbf{g}^u\| = 0$ **then**
        Find the Pareto stationarity, stop training;
      **end if**
      **if** $\cos\beta \geq 0$ **then**
        $\alpha^m = \alpha^u = \frac{1}{2}$;
      **end if**
      Integrate gradient: $\mathbf{h} = 2\alpha^m\mathbf{g}^m + 2\alpha^u\mathbf{g}^u$;
      $\mathbf{h} = \underbrace{\mathbf{h}/\|\mathbf{h}\|}_{\text{Keep non-conflict direction}} \cdot \underbrace{\|\mathbf{g}^m + \mathbf{g}^u\|}_{\text{Recovered magnitude}}$;
      Update $\theta^k$ with $\mathbf{h}$.
    **end for**
  **end for**

---

is common to all losses and maintain the necessary magnitude during gradient integration. Hence we first solve the Pareto optimization problem of Equation 2, obtaining $\alpha^m$ and $\alpha^u$, which could provide a non-conflict direction after integration. Furthermore, we recover the magnitude of final gradient to the same scale as the case without specific gradient integration, in case of the influence for generalization.

**Theorem 1.** *If the sequence of training iteration generated by the proposed MMPareto method is infinite, it admits a subsequence that converges to a Pareto stationarity.*

Beyond that, we also analyze the convergence of proposed MMPareto method. As Theorem 1, our algorithm is expected to converge to a Pareto stationarity. Detailed proof is provided in Appendix D. Overall, our MMPareto method could provide innocent uni-modal assistance whose gradients are with non-conflict direction and generalization guaranteed magnitude. It is expected to effectively alleviate the imbalanced multi-modal learning problem, enhancing primary multi-modal learning.

## 4 Experiment

### 4.1 Dataset and experiment settings

**CREMA-D** (Cao et al., 2014) is an audio-visual dataset for emotion recognition, covering 6 usual emotions. **Kinetics Sounds** (Arandjelovic & Zisserman, 2017) is an audio-visual dataset containing 31 human action classes. **Colored-and-gray-MNIST** (Kim et al., 2019) is a synthetic dataset based on MNIST (LeCun et al., 1998). Each instance contains two kinds of images, a gray-scale and a

Table 1: **Comparison with imbalanced multi-modal learning methods where bold and underline represent the best and runner-up respectively.** * indicates that the uni-modal evaluation (Acc audio and Acc video) is obtained by fine-tuning a uni-modal classifier with frozen trained uni-modal encoder, since this method could not provide uni-modal prediction directly. This evaluation method borrows from Peng et al. (2022). ↓ indicates a performance drop compared with uniform baseline.

| Method | CREMA-D | | | Kinetics Sounds | | |
|---|---|---|---|---|---|---|
| | Acc | Acc audio | Acc video | Acc | Acc audio | Acc video |
| Audio only | - | 61.69 | - | - | 53.63 | - |
| Video only | - | - | **56.05** | - | - | 49.20 |
| One joint loss* | 66.13 | 59.27 | 36.56 | 64.61 | 52.03 | 35.47 |
| Uniform baseline | 71.10 | 63.44 | 51.34 | 68.31 | 53.20 | 40.55 |
| G-Blending | 72.01 | 60.62 (↓) | 52.23 | 68.90 | 52.11 (↓) | 41.35 |
| OGM* | 69.19 (↓) | 56.99 (↓) | 40.05 (↓) | 66.79 (↓) | 51.09 (↓) | 37.86 (↓) |
| Greedy* | 67.61 (↓) | 60.69 (↓) | 38.17 (↓) | 65.32 (↓) | 50.58 (↓) | 35.97 (↓) |
| PMR* | 66.32 (↓) | 59.95 (↓) | 32.53 (↓) | 65.70 (↓) | 52.47 (↓) | 34.52 (↓) |
| MMPareto | **75.13** | **65.46** | 55.24 | **70.13** | **56.40** | **53.05** |

(a) Direction conflict.  (b) Magnitude difference.  (c) Direction conflict.  (d) Magnitude difference.

Figure 3: **(a&c).** Cosine similarity between gradients of multi-modal and uni-modal loss in the audio encoder of Kinetics Sounds and video encoder of CREMA-D, respectively. **(b&d).** Gradient magnitude in the audio encoder of Kinetics Sounds and video encoder of CREMA-D, respectively.

monochromatic colored image. **ModelNet40** (Wu et al., 2015) is a dataset with 3D objects, covering 40 categories. This dataset could be used to classify these 3D objects based on the 2D views of their front-view and back-view data (Su et al., 2015). Data of all views is 2D images of a 3D object.

When not specified, ResNet-18 (He et al., 2016) is used as the backbone in experiments and models are trained from scratch. Uni-modal modal features are integrated with late fusion method. For the Transformer backbone, MBT (Nagrani et al., 2021), is used as the backbone. Specifically, for the Colored-and-gray MNIST dataset, we build a neural network with 4 convolution layers and 1 average pool layer as the encoder, like Fan et al. (2023) does. More details are provided in Appendix A.

## 4.2 GRADIENT CONFLICT AND MAGNITUDE DIFFERENCE IN MULTI-MODAL SCENARIOS

In this section, we empirically verify the direction conflict and magnitude difference between multi-modal and uni-modal gradient. Firstly, in Figure 1a and Figure 3 (a&c), we show the cosine similarity between gradients on the Kinetics Sounds and CREMA-D. Based on the results, the update direction of multi-modal and uni-modal gradient indeed have conflict, *i.e.,* negative cosine similarity, which potentially brings risk for the optimization of the corresponding shared uni-modal encoder. In addition, such conflicts often exist in the early training stage, disturbance in which stage could substantially harm the model ability (Liu et al., 2020). In addition, we also observe the magnitude difference between gradients. Figure 3 (b&d) show the gradient magnitude of multi-modal and uni-modal gradients. These results verify the analysis that multi-modal gradient often has a smaller magnitude than that of uni-modal one.

## 4.3 OBSERVATION OF CONVENTIONAL PARETO INTEGRATION

To empirically verify the properties and performance of conventional Pareto method in the context of multi-modal learning, we conduct experiments across different dataset. Based on the former theoretical analysis, conventional Pareto method tends to result in the final gradient after integration

Table 2: **Comparison with imbalanced multi-modal learning methods where bold and underline represent the best and runner-up respectively.** The network is transformer-based backbone.

| Method | CREMA-D | | | | Kinetics Sounds | | | |
| | from scratch | | with pretrain | | from scratch | | with pretrain | |
| | Acc | mAP | Acc | mAP | Acc | mAP | Acc | mAP |
|---|---|---|---|---|---|---|---|---|
| One joint loss | 44.96 | 43.51 | 66.69 | 69.79 | 42.51 | 42.62 | 68.30 | 73.85 |
| Uniform baseline | 45.30 | 45.37 | 69.89 | 75.08 | 43.31 | 43.09 | 69.40 | 74.32 |
| G-Blending | 46.38 | 45.46 | 69.91 | 74.67 | 44.69 | 45.35 | 69.41 | 74.34 |
| OGM-GE | 42.88 | 39.13 | 65.73 | 68.28 | 41.79 | 41.02 | 69.55 | 74.46 |
| Greedy | 44.49 | 43.19 | 66.67 | 69.59 | 43.31 | 43.62 | 69.62 | 74.57 |
| PMR | 44.76 | 43.59 | 65.59 | 69.20 | 43.75 | 44.31 | 69.67 | 74.59 |
| MMPareto | **48.66** | **46.77** | **70.43** | **75.22** | **45.20** | **46.79** | **70.28** | **74.65** |

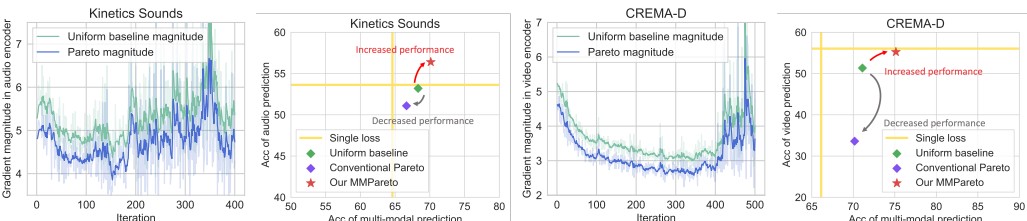

(a) Magnitude degradation.  (b) Methods comparison.  (c) Magnitude degradation.  (d) Methods comparison.

Figure 4: **(a&c).** Final gradient magnitude of Pareto method and uniform baseline in audio encoder of Kinetics Sounds and video encoder of CREMA-D. **(b&d).** Methods performance of multi-modal and uni-modal prediction in audio encoder of Kinetics Sounds and video encoder of CREMA-D.

having a degradation in magnitude compared with the case without any gradient integration strategy. Accordingly, the model generalization ability is potentially weakened. As shown in Figure 4 (a&c), the results indicate that after applying conventional Pareto method, there indeed exists a magnitude degradation in the final integrated gradient. What's more, based on the results in Figure 1c and Figure 4 (b&d), the conventional Pareto method loses its efficacy in multi-modal scenarios, even is clear worse than the uniform baseline, where all gradients are equally summed. These results demonstrate that conventional Pareto method does not fit well in the context of multi-modal learning. In contrast, our MMPareto, inspired by our theoretical finding, can handle this multi-modal case and provide both advanced multi-modal and uni-modal performance.

## 4.4 COMPARISON WITH IMBALANCED MULTI-MODAL LEARNING METHODS

To validate the effectiveness of our MMPareto method in overcoming imbalanced multi-modal learning problems, we compare it with recent studies: G-Blending (Wang et al., 2020), OGM-GE (Peng et al., 2022), Greedy (Wu et al., 2022) and PMR (Fan et al., 2023). One joint loss is the method that only uses multi-modal joint loss. And uniform baseline is the method in which multi-modal and uni-modal gradients are equally summed. To comprehensively evaluate the model ability, we further observe the uni-modal performance, besides the common multi-modal performance. Based on Table 1, we can find that the uniform baseline can achieve considerable performance, and even could outperform or be comparable with these imbalanced multi-modal learning methods. The reason could be that the introduction of uni-modal loss effectively enhances the learning of each modality, which accords with the core idea of these compared methods. Moreover, our MMpareto method with a conflict-free and generalization-guaranteed optimization process achieves a considerable improvement, compared with existing methods at the multi-modal prediction. More than that, our MMPareto method simultaneously exhibits outstanding uni-modal performance, and even can outperform solely trained uni-modal model. For example, Audio accuracy of MMPareto is superior to Audio-only method on both CREMA-D and Kinetics Sounds dataset.

Besides the CNN backbone, we also conduct experiments under the widely used Transformer backbone. The used backbone is MBT (Nagrani et al., 2021), which contains both single-modal layers as well as cross-modal interaction layers. Compared to the former CNN backbone with the late fusion method, uni-modal features in this transformer-based framework are more fully interacted and integrated. During experiments, we conduct experiments both from scratch and with

Table 3: **Comparison with related multi-task methods on Colored-and-gray-MNIST, Model-Net40 and Kinetics Sounds.** Bold and underline represent the best and runner-up respectively.

| Method | CG-MNIST (Color/Gray) | | ModelNet40 (Front/Back View) | | Kinetics Sounds (Audio/Video) | |
|---|---|---|---|---|---|---|
| | Acc | mAP | Acc | mAP | Acc | mAP |
| One joint loss | 60.50 | 60.43 | 87.88 | 77.36 | 64.61 | 57.62 |
| Uniform baseline | 75.68 | 77.66 | 89.18 | 82.75 | 68.31 | 61.28 |
| GradNorm | 76.16 | 78.68 | 88.98 | 82.35 | 65.84 | 60.29 |
| PCGrad | 79.35 | 82.78 | 89.59 | 83.57 | 69.11 | 65.07 |
| MetaBalance | 79.18 | 81.87 | 89.63 | 83.03 | 68.90 | 64.31 |
| MMPareto | **81.88** | **84.34** | **89.95** | **85.09** | **70.13** | **68.44** |

ImageNet pre-training. Results are shown in Table 2. Based on the results, we can have the following observation. Firstly, former imbalanced multi-modal learning could lose efficacy under these more complex scenarios with cross-modal interaction. For example, OGM-GE method is even worse than the one joint loss method on CREMA-D dataset. In contrast, our MMPareto gradient integration strategy is not only applicable to CNN backbones, but also able to maintain superior performance in transformer-based frameworks with complex interactions. In addition, whether or not to use pre-training does not affect the effectiveness of the method, which reflects its flexibility.

### 4.5 COMPARISON WITH RELATED MULTI-TASK METHODS

In past studies, there are other strategies that are used to balance multiple learning objectives. In this section, we compare several representative ones: GradNorm (Chen et al., 2018), PCGrad (Yu et al., 2020), MetaBalance (He et al., 2022). Experiments are conducted on different multi-modal dataset, covering six kinds of modalities. Based on the results in Table 3, we can conclude that former multi-task methods are also possibly invalid in the context of multi-modal learning. For example, GradNorm method is inferior to the uniform baseline on both ModelNet40 and Kinetics Sounds dataset. In contrast, our MMPareto method, which specifically considers the multi-modal property that there is a magnitude discrepancy between multi-modal and uni-modal gradient, maintains its superior performance across various dataset with different kinds of modalities.

### 4.6 EXTENSION TO MULTI-TASK SCENARIO

To evaluate scalability of our method in multi-task cases with similar property that there is a clear discrepancy in task difficulty, we conduct experiments on MultiMNIST dataset (Sabour et al., 2017). In MultiMNIST, two images with different digits from the original MNIST dataset are picked, and then combined into a new one by putting one digit on the left and the other one on the right. Two tasks are to classify these two digits. In order to increase the difference in difficulty of tasks, we add 50% salt-and-pepper noise on the right part of images. We provide data samples in Appendix B. Based on Table 4, conventional Pareto method also fails under this multi-task case. Not surprisingly, our MMPareto could extend to this scenario and achieve considerable performance, indicating its ideal scalability.

Table 4: Results on MultiMNIST with 50% salt-and-pepper noise on the right part of images.

| Method | Accuracy | |
|---|---|---|
| | Task 1 | Task2 |
| Uniform baseline | 86.63 | 78.42 |
| Conventional Pareto | 86.95 | 77.04 ($\downarrow$) |
| MMPareto | **87.72** | **80.64** |

## 5 CONCLUSION

In this paper, we first identify the potential gradient conflict in the multitask-like framework for the imbalanced multi-modal learning problem. To solve conflicts, the idea of Pareto integration is introduced and theoretically analyzed in the context of multi-modal learning. Then, we propose MMPareto method, which can provide innocent uni-modal assistance with diminished conflict and guaranteed generalization for multi-modal learning, effectively alleviating the imbalance multi-modal learning problem. Furthermore, this method could also extend to multi-task cases with a clear discrepancy in task difficulty, indicating its scalability.

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

## A  DATASET AND EXPERIMENT SETTINGS

**CREMA-D** (Cao et al., 2014) is an audio-visual dataset for emotion recognition, including 7,442 video clips, each spanning 2 to 3 seconds in duration. The video content is that actors speak several short words. This dataset covers 6 emotions: angry, happy, sad, neutral, discarding, disgust and fear.

**Kinetics Sounds** (Arandjelovic & Zisserman, 2017) is an audio-visual dataset containing 31 human action classes selected from Kinetics dataset (Kay et al., 2017). All videos are manually annotated for human action using Mechanical Turk and cropped to 10 seconds long around the action.

**Colored-and-gray-MNIST** (Kim et al., 2019) is a synthetic dataset based on MNIST (LeCun et al., 1998). Each instance contains two kinds of images, a gray-scale and a monochromatic colored image. Monochromatic images in the training set are strongly color-correlated with their digit labels, while monochromatic images in the other sets are weakly color-correlated with their labels.

**ModelNet40** (Wu et al., 2015) is a dataset with 3D objects, covering 40 categories. It contains 9,483 training samples and 2,468 test samples. This dataset could be used to classify these 3D objects based on the 2D views of their front-view and back-view data (Su et al., 2015). Data of all views is a collection of 2D images of a 3D object.

When not specified, ResNet-18 (He et al., 2016) is used as the backbone in experiments and models are trained from scratch. Concretely, for the visual encoder, we take multiple frames as the input, and feed them into the 2D network like Zhao et al. (2018) does; for the audio encoder, we modified the input channel of ResNet-18 from three to one like Chen et al. (2020) does and the rest parts remain unchanged; Encoders of other modalities are not modified. For the CNN backbone, we use the widely used late fusion method, to integrate uni-modal features. For the Transformer backbone, MBT (Nagrani et al., 2021), is used as the backbone. Concretely, the backbone contains 6 single-modal layers and 2 layers with cross-modal interaction. Specifically, for the Colored-and-gray MNIST dataset, we build a neural network with 4 convolution layers and 1 average pool layer as the encoder, like Fan et al. (2023) does. During the training, we use SGD with momentum (0.9) and set the learning rate at $1e-3$. All models are trained on 2 NVIDIA RTX 3090 (Ti).

## B  SAMPLES OF MULTIMNIST DATASET

Here we provide several samples of MultiMNIST dataset. In MultiMNIST, two images with different digits from the original MNIST dataset are picked, and then combined into a new one by putting one digit on the left and the other one on the right. Two tasks are to classify these two digits. In order to increase the difference in difficulty between tasks, we add $50\%$ salt-and-pepper noise on the right part of images.

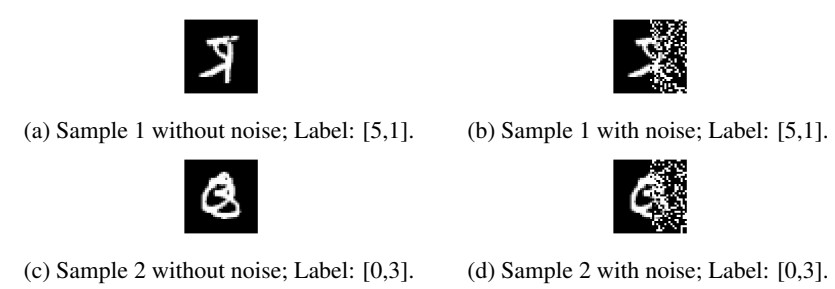

(a) Sample 1 without noise; Label: [5,1].  (b) Sample 1 with noise; Label: [5,1].

(c) Sample 2 without noise; Label: [0,3].  (d) Sample 2 with noise; Label: [0,3].

Figure 5: Samples in MultiMNIST dataset.

## C  PROOF FOR GENERALIZATION OF CONVENTIONAL PARETO METHOD

### C.1  THE PAC-BAYESIAN FRAMEWORK

The PAC-Bayesian theorem provides a generalization bound on randomized classifiers. Suppose the prior distribution over the parameter space $\Theta$ is $P$. Let $Q$ is the distribution on the parameter space $\Theta$

expressing the learnt hypothesis function. Denote the expected risk as $\mathcal{R}(Q)$ and empirical risk as $\hat{\mathcal{R}}(Q)$. Then, a classic result uniformly bounding the expected risk $\mathcal{R}(Q)$ in terms of the empirical risk $\hat{\mathcal{R}}$ and the KL divergence $\mathcal{D}(Q\|P)$ is as follows.

**Lemma 1** ((McAllester, 1999), Theorem 1). *For any positive real $\delta \in (0, 1)$, with probability as least $1 - \delta$ over a sample set of size $N$, we have the following inequality for all distributions $Q$:*

$$\mathcal{R}(Q) \leq \hat{\mathcal{R}}(Q) + \sqrt{\frac{\mathcal{D}(Q\|P) + \log\frac{1}{\delta} + \frac{5}{2}\log N + 8}{2N - 1}}, \tag{6}$$

*where $\mathcal{D}(Q\|P) = \mathbb{E}_{\theta \sim Q}\left(\log\frac{Q(\theta)}{P(\theta)}\right)$.*

## C.2 SGD AS ORNSTEIN-UHLENBECK PROCESS

**Lemma 2** ((Mandt et al., 2017), Appendix B). *Under the 2-order differentiable assumption, SGD can be translated as Ornstein-Uhlenbeck process with stationary distribution:*

$$q(\theta) \propto \exp\left\{-\frac{1}{2}\theta^\top \Sigma^{-1}\theta\right\}, \tag{7}$$

*where the covariance $\Sigma$ satisfies:*

$$A\Sigma + \Sigma A = \frac{\eta}{|S|}C. \tag{8}$$

$A$ is the Hessian at the optimum and $C$ is the covariance matrix, bringing by the random sampling within SGD.

## C.3 SGD GENERALIZATION BOUND OF MULTITASK-LIKE MULTI-MODAL FRAMEWORK

Formally, machine learning algorithms are designed to learn a hypothesis function with the lowest expected risk $\mathcal{R}$ under the loss function $\mathcal{L}$ from a hypothesis class $F_\theta$, where $\theta$ is the parameter of the hypothesis. Suppose the parameter follows a distribution $Q$, the expected risks respectively in terms of $\theta$ and $Q$ are defined as:

$$\begin{aligned} \mathcal{R}(\theta) &= \mathbb{E}_{(X,Y)\sim\mathcal{D}}\mathcal{L}\left(F_\theta(X), Y\right), \\ \mathcal{R}(Q) &= \mathbb{E}_{\theta\sim Q}\mathbb{E}_{(X,Y)\sim\mathcal{D}}\mathcal{L}\left(F_\theta(X), Y\right), \end{aligned} \tag{9}$$

where $(X, Y) \sim \mathcal{D}$ is the training data. Since the data distribution $\mathcal{D}$ is not available. In practice, it is often substituted with the empirical risk $\hat{\mathcal{R}}$:

$$\begin{aligned} \hat{\mathcal{R}}(\theta) &= \frac{1}{|N|}\sum_{i=1}^{N}\mathcal{L}\left(F_\theta\left(X_i\right), Y_i\right), \\ \hat{\mathcal{R}}(Q) &= \mathbb{E}_{\theta\sim Q}\left[\frac{1}{N}\sum_{i=1}^{N}\mathcal{L}\left(F_\theta\left(X_i\right), Y_i\right)\right], \end{aligned} \tag{10}$$

where $N$ is the number of training samples and $(X_i, Y_i)$ is the $i$-th sample within them.

To optimize the expected risk, gradient descent (GD) is the often used method. Specifically, suppose the training set is with $N$ samples. The gradient in item of parameter $\theta$ is as follows:

$$\mathbf{g}_N(\theta(t)) = \nabla_{\theta(t)}\hat{\mathcal{R}}(\theta(t)) = \frac{1}{N}\sum_{i=1}^{N}\nabla_{\theta(t)}\mathcal{L}\left(F_{\theta(t)}\left(X_i\right), Y_i\right), \tag{11}$$

where $\theta(t)$ is the parameter at iteration $t$. $\mathbf{g}_N(\theta(t))$ is considered as the full gradient over all $N$ training samples at iteration $t$.

In practice, Stochastic gradient descent (SGD) is more widely used to optimize the network. In SGD, it estimates the full gradient based on mini-batches of the training samples. Here we denote $S$ be a

set of mini-batch. All mini-batchs are independently and identically drawn from the training samples. Then, the gradient of SGD for the mini batch $S$ at iteration $t$ is:

$$\mathbf{g}_S(\theta(t)) = \frac{1}{|S|} \sum_{n \in S} \nabla_{\theta(t)} \mathcal{L} \left( F_{\theta(t)} \left( X_n \right), Y_n \right), \tag{12}$$

Based on former studies of SGD optimization (Jastrzebski et al., 2017), gradient of a mini-batch $\mathbf{g}_S(\theta(t))$ is un-biased estimations of full gradient $\mathbf{g}_N(\theta(t))$. And for a sufficiently large batch size, based on the central limit theorem, the gradient of each mini-batch satisfies:

$$\mathbf{g}_S(\theta(t)) \sim \mathcal{N} \left( \mathbf{g}_N(\theta(t)), \frac{1}{|S|} C \right), \tag{13}$$

where $C$ is the covariance matrix, bringing by the random sampling.

With this property of SGD optimization, we can further consider the multitask-like multi-modal framework, with multi-modal loss function $\mathcal{L}_m$ and uni-modal loss function $\mathcal{L}_u^k$. For $\theta^k$, the uni-modal encoder parameter of modality $k$, gradients of $\mathcal{L}_m$ and $\mathcal{L}_u^k$ at iteration $t$ satisfy:

$$\begin{aligned} \mathbf{g}_S^m(\theta^k(t)) &\sim \mathcal{N} \left( \mathbf{g}_N^m(\theta^k(t)), \frac{1}{|S|} C^m \right), \\ \mathbf{g}_S^u(\theta^k(t)) &\sim \mathcal{N} \left( \mathbf{g}_N^u(\theta^k(t)), \frac{1}{|S|} C^u \right), \end{aligned} \tag{14}$$

where $C^m$ and $C^u$ are the covariance matrix for $\mathcal{L}_m$ and $\mathcal{L}_u^k$, respectively. During training, $\mathcal{L}_m$ and $\mathcal{L}_u^k$ are calculated and back-propagation independently, so their gradients can be considered as be independent. Then, when without any gradient integration strategy (uniform baseline where all losses are equally summed), the final gradient is:

$$\begin{aligned} \mathbf{h}_S(\theta^k(t)) &= \mathbf{g}_S^m(\theta^k(t)) + \mathbf{g}_S^u(\theta^k(t)), \\ \mathbf{h}_S(\theta^k(t)) &\sim \mathcal{N} \left( \mathbf{g}_N^m(\theta^k(t)) + \mathbf{g}_N^u(\theta^k(t)), \frac{C^m + C^u}{|S|} \right). \end{aligned} \tag{15}$$

Then, use the final gradient $\mathbf{h}_S(\theta^k(t))$ to iteratively update the parameter $\theta^k$:

$$\begin{aligned} \theta^k(t+1) &= \theta^k(t) - \eta \mathbf{h}_S(\theta^k(t)), \\ \theta^k(t+1) &= \theta^k(t) - \eta(\mathbf{g}_S^m(\theta^k(t)) + \mathbf{g}_S^u(\theta^k(t))), \\ \theta^k(t+1) &= \theta^k(t) - \eta(\mathbf{g}_N^m(\theta^k(t)) + \mathbf{g}_N^u(\theta^k(t))) + \eta \epsilon_t, \\ \theta^k(t+1) &= \theta^k(t) - \eta \mathbf{h}_N(\theta^k(t)) + \eta \epsilon_t, \end{aligned} \tag{16}$$

where $\epsilon_t \sim \mathcal{N} \left( 0, \frac{C^m + C^u}{|S|} \right)$ and $\eta > 0$ is the learning rate. $\mathbf{h}_N(\theta^k(t)) = \mathbf{g}_N^m(\theta^k(t)) + \mathbf{g}_N^u(\theta^k(t))$.

For small enough constant learning rate, SGD can be treated as the numerical discretization of the following stochastic differential equation (SDE) (Li et al., 2017; Mandt et al., 2017), which is a Ornstein-Uhlenbeck process:

$$d\theta^k = -\mathbf{h}_N(\theta^k)dt + \sqrt{\frac{\eta}{|S|}} B dW(t), \tag{17}$$

where $W(t)$ is a while noise and follows $\mathcal{N}(0, I)$ and $C^m + C^u = BB^\top$. As $C^m$ and $C^u$ can both considered as symmetric positive-semidefinite matrix, $C^m + C^u$ can be factorized as $BB^\top$. This assumption has been primarily used in the former theoretically analysis (Mandt et al., 2017).

Moreover, assuming that the loss function in the local region around the minimum is convex and 2-order differentiable, based on Lemma 2, this Uhlenbeck process has an analytic stationary distribution $q(\theta^k)$ that is Gaussian:

$$q(\theta^k) \propto \exp \left\{ -\frac{1}{2} \theta^{k^\top} \Sigma^{-1} \theta^k \right\}, \tag{18}$$

where the covariance $\Sigma$ satisfies:

$$\Sigma A + A\Sigma = \frac{\eta}{|S|} BB^\top = \frac{\eta}{|S|}(C^m + C^u). \tag{19}$$

$A$ is the Hessian matrix around the optimum.

To further build the generalization bound of SGD, we utilize the PAC-Bayesian framework, inspired by He et al. (2019). Therefore, based on Lemma 1, it is essential to build the KL divergence between learnt distribution and the prior distribution. The learnt distribution over $\theta^k$ of SGD has been provided in Equation 18. In addition, the prior distribution can be interpreted as the distribution of initial parameters. In practice, the parameters are often initialized as Gaussian distribution or uniform distributions. Therefore, we use a standard Gaussian distribution $\mathcal{N}(0, I)$ as the prior distribution. Then we can have the densities of learnt distribution $Q$ and prior distribution $P$ is:

$$
\begin{aligned}
q(\theta^k) &= \frac{1}{\sqrt{2\pi \det(\Sigma)}} \exp\left\{ -\frac{1}{2}\theta^{k\top}\Sigma^{-1}\theta^k \right\}, \\
p(\theta^k) &= \frac{1}{\sqrt{2\pi \det(I)}} \exp\left\{ -\frac{1}{2}\theta^{k\top}I\theta^k \right\}.
\end{aligned}
\tag{20}
$$

Denote $\Theta^k = \mathbb{R}^d$ as the parameter space of $\theta^k$. Then we can have the KL divergence between distribution $Q$ and $P$:

$$
\begin{aligned}
&\mathcal{D}(Q\|P) \\
=&\mathbb{E}_{\theta^k \sim Q}\left( \log \frac{Q(\theta^k)}{P(\theta^k)} \right) \\
=&\int_{\theta^k \in \Theta^k} \log\left( \frac{q(\theta^k)}{p(\theta^k)} \right) q(\theta^k)\mathrm{d}\theta^k \\
=&\int_{\theta^k \in \Theta^k} \left[ \frac{1}{2}\log\left( \frac{1}{\det(\Sigma)} \right) + \frac{1}{2}\left( \theta^{k\top}I\theta^k - \theta^{k\top}\Sigma^{-1}\theta^k \right) \right] q(\theta^k)\mathrm{d}\theta^k \\
=&\frac{1}{2}\log\left( \frac{1}{\det(\Sigma)} \right) + \frac{1}{2}\operatorname{tr}(\Sigma - I).
\end{aligned}
\tag{21}
$$

Since the parameter dimension is $d$, we know that trace of $I$ is $d$. For $\Sigma$, based on Equation 19:

$$
\begin{aligned}
\Sigma A + A\Sigma &= \frac{\eta}{|S|}(C^m + C^u) \\
\frac{\eta}{|S|}(C^m + C^u)A^{-1} &= A\Sigma A^{-1} + \Sigma \\
\operatorname{tr}\left( \frac{\eta}{|S|}(C^m + C^u)A^{-1} \right) &= \operatorname{tr}\left( A\Sigma A^{-1} + \Sigma \right) \\
\operatorname{tr}\left( \frac{\eta}{|S|}(C^m + C^u)A^{-1} \right) &= \operatorname{tr}\left( \Sigma A^{-1}A \right) + \operatorname{tr}(\Sigma) \\
\operatorname{tr}\left( \frac{\eta}{|S|}(C^m + C^u)A^{-1} \right) &= 2\operatorname{tr}(\Sigma)
\end{aligned}
\tag{22}
$$

Then, we can have:

$$\operatorname{tr}(\Sigma) = \frac{1}{2}\operatorname{tr}\left( \frac{\eta}{|S|}(C^m + C^u)A^{-1} \right) = \frac{1}{2}\frac{\eta}{|S|}\operatorname{tr}\left( (C^m + C^u)A^{-1} \right). \tag{23}$$

Then, with $\operatorname{tr}(\Sigma)$ and $\operatorname{tr}(I)$, we have:

$$
\begin{aligned}
&\mathcal{D}(Q\|P) \\
=&\frac{1}{2}\log\left( \frac{1}{\det(\Sigma)} \right) + \frac{1}{2}\operatorname{tr}(\Sigma - I) \\
=&\frac{1}{4}\frac{\eta}{|S|}\operatorname{tr}\left( (C^m + C^u)A^{-1} \right) - \frac{1}{2}d - \frac{1}{2}\log(\det(\Sigma)).
\end{aligned}
\tag{24}
$$

Denote the expected risk as $\mathcal{R}(Q)$ and empirical risk as $\hat{\mathcal{R}}(Q)$. Then, combine Equation 24 with the PAC-Bayesian generalization bound, for any positive real $\sigma \in (0, 1)$, with probability at least $1 - \sigma$ over a training sample size of size $N$, we have:

$$\mathcal{R}(Q) \leq \hat{\mathcal{R}}(Q) + \sqrt{\frac{\frac{\eta}{|S|} \text{tr}\left((C^m + C^u)A^{-1}\right) - 2d - 2\log(\det(\Sigma)) + 4\log\left(\frac{1}{\delta}\right) + 10\log N + 32}{8N - 4}}.$$
(25)

Overall, we now have the SGD generalization for uniform baseline case, where no gradient integration methods are applied.

### C.4 INFLUENCE OF CONVENTIONAL PARETO FOR SGD GENERALIZATION

Here we further consider the SGD optimization with conventional Pareto gradient integration. We know that at each iteration, Pareto method would return weight $\alpha^m$ and $\alpha^u$ for the integration of $\mathbf{g}_S^m(\theta^k)$ and $\mathbf{g}_S^u(\theta^k)$. Then, when updateing parameter, we have:

$$\theta^k(t+1) = \theta^k(t) - \eta\mathbf{h}^{\text{Pareto}}{}_S(\theta^k),$$
$$\theta^k(t+1) = \theta^k(t) - \eta(2\alpha^m\mathbf{g}_S^m(\theta^k) + 2\alpha^u\mathbf{g}_S^u(\theta^k)).$$
(26)

As stated in the manuscript, we use $2\alpha^i$ as the gradient weight is to keeps the same weight summation with uniform baseline, where all gradient weight is assigned as 1 and no gradient integration methods are applied. When applying Pareto integration, both the magnitude and direction of $\mathbf{h}^{\text{Pareto}}{}_S(\theta^k)$ are different from uniform baseline.

Based on the former analysis, for the uniform baseline case without any gradient integration method, we have:

$$\mathbf{h}_S(\theta^k(t)) = \mathbf{g}_S^m(\theta^k(t)) + \mathbf{g}_S^u(\theta^k(t)),$$
$$\mathbf{h}_S(\theta^k(t)) \sim \mathcal{N}\left(\mathbf{g}_N^m(\theta^k(t)) + \mathbf{g}_N^u(\theta^k(t)), \frac{C^m + C^u}{|S|}\right).$$
(27)

Since gradient weight under any case is non-negative (both uniform baseline and Pareto integration), the final gradient after Pareto integration is in the convex hull of the family of gradient vectors $\{\{\mathbf{g}_S^i(\theta^k(t))\}_{i\in\{m,u\}}$. Then, we can find a new convex combination $2\overline{\alpha^m}\mathbf{g}_S^m(\theta^k(t)) + 2\overline{\alpha^m}\alpha^u\mathbf{g}_S^u(\theta^k(t))$, which have the same direction with Pareto direction $(2\alpha^m\mathbf{g}_S^m(\theta^k(t)) + 2\alpha^u\mathbf{g}_S^u(\theta^k(t)))$ and the same magnitude with the uniform baseline $(\mathbf{g}_S^m(\theta^k(t)) + \mathbf{g}_S^u(\theta^k(t)))$. Since the gradient vector follows random distribution in SGD, its direction itself could oscillate, here we mainly consider the influence of gradient magnitude. So we can view $2\overline{\alpha^m}\mathbf{g}_S^m(\theta^k(t)) + 2\overline{\alpha^m}\alpha^u\mathbf{g}_S^u(\theta^k(t))$ and uniform baseline with the same gradient magnitude case having the same distribution:

$$2\overline{\alpha^m}\mathbf{g}_S^m(\theta^k(t)) + 2\overline{\alpha^m}\alpha^u\mathbf{g}_S^u(\theta^k(t)) \sim \mathcal{N}\left(\mathbf{g}_N^m(\theta^k(t)) + \mathbf{g}_N^u(\theta^k(t)), \frac{C^m + C^u}{|S|}\right).$$
(28)

Then, we can have:

$$\mathbf{h}^{\text{Pareto}}{}_S(\theta^k(t)) = \lambda \cdot (2\overline{\alpha^m}\mathbf{g}_S^m(\theta^k(t)) + 2\overline{\alpha^m}\alpha^u\mathbf{g}_S^u(\theta^k(t))),$$
(29)

where $\lambda$ can considered as the magnitude difference between Pareto method and the uniform baseline case without any specifically designed gradient integration. Based on Remark 1, the Pareto integration tends to assign larger weight for gradient with smaller magnitude. Accordingly, the magnitude of final gradient of Pareto method is smaller than that of uniform baseline case. Therefore, we have that $0 < \lambda < 1$ at each iteration.

Since the magnitude difference between Pareto method and the uniform baseline case is changing during training, the concrete value of $\lambda$ is accordingly changing. Therefore, we use its largest value $\gamma$

which is constant, to substitute for it. Then, the parameter update equation with Pareto integration is:

$$\theta^k(t+1) = \theta^k(t) - \eta \mathbf{h}^{\text{Pareto}}(\theta^k),$$
$$\theta(t+1)^k = \theta^k(t) - \eta\gamma(2\overline{\alpha^m}\hat{\mathbf{g}}_S^m(\theta^k(t)) + 2\overline{\alpha^m}\alpha^u\hat{\mathbf{g}}_S^u(\theta^k(t))),$$
$$\theta^k(t+1) = \theta^k(t) - \eta\gamma(\mathbf{g}_N^m(\theta^k(t)) + \mathbf{g}_N^u(\theta^k(t))) + \eta\gamma\epsilon_t,$$
$$\theta^k(t+1) = \theta^k(t) - \eta\gamma\mathbf{h}_N(\theta^k(t)) + \eta\gamma\epsilon_t,$$

(30)

where $\epsilon_t \sim \mathcal{N}\left(0, \frac{C^m+C^u}{|S|}\right)$.

Similarly, for small enough constant learning rate, SGD with Pareto integration can be treated as the numerical discretization of the following stochastic differential equation (SDE) Li et al. (2017); Mandt et al. (2017), which is a Ornstein-Uhlenbeck process:

$$d\theta^k = -\mathbf{h}_N(\theta^k)dt + \sqrt{\frac{\eta\gamma}{|S|}}BdW(t),$$

(31)

where $W(t)$ is a while noise and follows $\mathcal{N}(0, I)$ and $C^m + C^u = BB^\top$. As $C^m$ and $C^u$ can both considered as symmetric positive-semidefinite matrix, $C^m + C^u$ can be factorized as $BB^\top$. This assumption has been primarily used in the former theoretically analysis (Mandt et al., 2017).

Following the analysis process as Section C.3, we can have the generalization bound for SGD with Pareto integration. For any positive real $\sigma \in (0, 1)$, with probability at least $1 - \sigma$ over a training sample size of size $N$, we have:

$$\mathcal{R}(Q) \le \hat{\mathcal{R}}(Q) + \sqrt{\frac{\frac{\eta\gamma}{|S|}\operatorname{tr}\left((C^m + C^u)A^{-1}\right) - 2d - 2\log(\det(\Sigma)) + 4\log\left(\frac{1}{\delta}\right) + 10\log N + 32}{8N - 4}}$$

(32)

Based on the above analysis, we know that $0 < \gamma < 1$. Compared with Equation 25, the upper bound is decreased. Therefore, the generalization ability of SGD with Pareto integration is weakened, compared with the case without any specifically designed gradient integration. Moreover, here $\gamma$ is the largest value of $\lambda$, *i.e.,* the least gradient magnitude decrease value of Pareto integration, during training. Hence this is in fact a loose bound. Model generalization would be affected more in practice.

## D    PROOF FOR THE CONVERGENCE OF MMPARETO

**Theorem 1.** *If the sequence of training iteration generated by the proposed MMPareto method is infinite, it admits a subsequence that converges to a Pareto stationarity.*

*Proof.* In MMPareto algorithm, at each training iteration, we first sovle the optimization problem:

$$\min_{\alpha^m, \alpha^u \in \mathcal{R}} \|\alpha^m \mathbf{g}^m + \alpha^u \mathbf{g}^u\|^2$$
$$s.t. \quad \alpha^m, \alpha^u \ge 0, \alpha^m + \alpha^u = 1.$$

(33)

For brevity, here we use $\{\mathbf{g}^i\}_{i \in \{m,u\}}$ to substitute $\{\mathbf{g}_S^i(\theta^k(t))\}_{i \in \{m,u\}}$. $\|\cdot\|$ denotes the $L_2$-norm. This problem is equal to find the minimum-norm in the convex hull of the family of gradient vectors $\{\mathbf{g}^i\}_{i \in \{m,u\}}$. We denote the found minimum-norm as $\omega = \alpha^m \mathbf{g}^m + \alpha^u \mathbf{g}^u$. Based on Désidéri (2012), either $\omega$ to this optimization problem is 0 and the corresponding parameters is Pareto-stationary which is a necessary condition for Pareto-optimality, or $\omega$ can provide a descent directions common to all learning objectives. When the minimum-norm $\omega$ does not satisfy the condition of Pareto stationarity, we consider the non-conflict case and conflict case respectively.

We first analyze the non-conflict case, where $\cos\beta \ge 0$. $\beta$ is the angle between $\mathbf{g}^m$ and $\mathbf{g}^u$. Under this case, the arbitrary convex combination of the family of gradient vectors $\{\mathbf{g}^i\}_{i \in \{m,u\}}$ is common to all learning objectives. Therefore, to maintain the gradient magnitude as the case without specific gradient integration, we assign $\alpha^m = \alpha^u = \frac{1}{2}$. Then, the final gradient $\mathbf{h} = 2\alpha^m \mathbf{g}^m + 2\alpha^u \mathbf{g}^u$ is with direction that can benefit all losses and generalization guaranteed magnitude.

Then we analyze the conflict case, where $\cos\beta < 0$. The results of optimization problem Equation 33 are used as $\alpha^m$ and $\alpha^u$. Based on above statement, we can have that $\mathbf{h} = 2\alpha^m \mathbf{g}^m + 2\alpha^u \mathbf{g}^u$

can provide a direction that is common to all learning objective. Furthermore, we recover the magnitude of final gradient to the same scale with the uniform baseline, to avoid potential risk for model generalization. Overall, the final gradient can also benefit all losses and have generalization guaranteed magnitude.

In summary, the final gradient of MMPareto could always provide the direction that is common to all learning objectives. If the training iteration stops in a finite number of steps, a Pareto-stationary point has been reached. Otherwise, the iteration continues indefinitely, generating an infinite sequence of shared parameters $\theta^k$. Since the value of loss function $\mathcal{L}_m$ and $\mathcal{L}_u^k$ is positive and monotone-decreasing during optimization, it is bounded. Hence, the sequence of parameter $\theta^k$ is itself bounded and it admits a subsequence converging to $\theta^{k^*}$.

Necessarily, $\theta^{k^*}$ is a Pareto-stationary point. In other words, the minimum-norm $\omega^*$ is zero at this step. To establish this, assume instead that the obtained minimum-norm $\omega^*$, which corresponds to $\theta^{k^*}$, is nonzero. A new iteration would potentially diminish each learning objective of a finite amount, and a better solution of parameter $\theta^k$ be found.

Therefore, if the sequence of training iteration generated by the proposed MMPareto method is infinite, it admits a subsequence that converges to a Pareto stationarity.

$\square$

