# OpenReview forum: "MMPareto: Innocent Uni-modal Assistance for Enhanced Multi-modal Learning"
_ICLR.cc/2024/Conference — ICLR 2024 Conference Withdrawn Submission_

### Official Review · Reviewer_SyiT · 2023-10-23

**Soundness:** 2 fair
**Presentation:** 3 good
**Contribution:** 2 fair
**Rating:** 3
**Confidence:** 4

**Summary:**

This paper focus on enhancing the efficacy of multimodal-learning. Specifically, the authors claim that conflict between multimodal and unimodal learning objectives may be harmful to the performance of uni-modal encoders. To address this issue, the authors propose to intergrate the gradients of unimodal and multimodal losses using a modified Pareto method. The authors also try to developed a theory to analysis the benefit of the proposed method. They claim that the established generalization bound for conventional Pareto is looser than its counterparts. Experimental studies are taken to demonstrate the effectiveness of the proposed MMPareto.

**Strengths:**

- Imbalanced multi-modal learning is an interesting topic and important for multimodal data with unbalanced quality.
- It is appreciated that the authors present theoretical results trying to reveal why conventional Pareto integration methods fails. Given the theorical analysis, I think the proposed method is well motivated.
- According to the empirical results, it seems that the proposed method mitigate inbalanced multimodal learning problem and extensive empirical results validate the effectiveness.
- The paper follows a good construction. I enjoy the authors' neat and clear writing.

**Weaknesses:**

Major concerns:
- There seems to be some ambiguities in the theoretical results. The established generalization error bounds of SGD without any gradient integration strategy and Pareto gradient integration are shown in equation. 4 and equation. 5 respectively. As the authors claim, we have $0\leq\gamma\leq 1$ for Pareto gradient integration. It seems that the generalization error bound of Pareto gradient integration (i.e., eq. 5) is not loose but tighter than that in eq. 4. This is very confusing for me, and thus raise my concern about the soundness of the main theoretical results.
- Although the authors try to shed some light upon the pitfall of conventional Pareto algorithm considering multimodal learning problem. Their theorical results may not support the proposed method MMPareto directly. Is the proposed MMPareto have a tighter generalization error bound than their counterparts? The authors should clarify this more explicitly.
- The proposed method is working under the condition that multitask-like learning object is involved. However, many multimodal learning models can not give a prediction on unimodal input. I think the authors need to clarify this limitation in the future.

**Questions:**

Please address the issues raised in the weakness section.

---

> ### Author Response · Authors · 2023-11-22
>
> Dear reviewer,
>
> Thank you for taking the time to review our paper. We greatly appreciate your insightful feedback and constructive comments.
>
> We thank you for pointing out the concern about the established generalization error bounds of SGD. We realize this issue and rethink the effect of generalization when using traditional Pareto method.
>
> One possible idea is that we know the magnitude will be decreased when using traditional Pareto method. Then, based on the past studies [A,B], the decrease in gradient magnitude in SGD will weaken the noise term in SGD, then accordingly affect the generalization.
>
> Once again, thank you for your valuable review and we will resubmit the revised manuscript for the next conference.
>
> Sincerely,
>
> Authors
>
>
>
> [A] Jastrzębski, S., Kenton, Z., Arpit, D., Ballas, N., Fischer, A., Bengio, Y., & Storkey, A. (2017). Three factors influencing minima in sgd. arXiv preprint arXiv:1711.04623.
>
> [B] Zhu, Z., Wu, J., Yu, B., Wu, L., & Ma, J. The Anisotropic Noise in Stochastic Gradient Descent: Its Behavior of Escaping from Sharp Minima and Regularization Effects. In ICML 2019.

---

### Official Review · Reviewer_45kN · 2023-10-31

**Soundness:** 2 fair
**Presentation:** 2 fair
**Contribution:** 2 fair
**Rating:** 5
**Confidence:** 3

**Summary:**

This paper tackles the problem of imbalanced multi-modal learning where all modalities are not utilized properly by the model leading to limited performance. The authors point out gradient conflicts between uni-modal and multi-modal losses as a potential reason for this problem. Noting the similarity to gradient conflicts that exist in multi-task learning, they propose to utilize the pareto integration strategy from the multi-task learning literature to obtain gradients which are common to both the uni-modal and multi-modal objectives. However, the multi-modal problem generally leads to smaller gradients for the multi-modal loss which leads to the conventional method to assign larger weight to only the multi-modal gradient. Thus, they propose to rescale the gradients to better the uni-modal performance. Experiments are conducted on two audio-visual datasets (CREMA-D and Kinetics Sounds) and Colored/Gray MNIST and achieves good performance in both uni-modal and multi-modal cases when compared to baselines.

**Strengths:**

1. Multi-modal learning is an important practical problem and in the real-world one or more modalities may be missing during inference, thus, it is crucial to ensure each individual uni-modal block is up to the mark. In this regard, the problem attempted by the paper is pretty significant.

2. The paper does a good job in building the connection between multi-modal and multi-task learning and modifying the pareto strategy to accordingly to account for the gradient norm differences between the multi-modal and uni-modal losses.

**Weaknesses:**

1. With contrastive learning becoming the state-of-the-art for multi-modal learning (e.g. CLIP), it is not clear how these strategies transfer or even if they are useful. A discussion would be useful.

2. Eq. 4,5 try to explain the generalization with the pareto integration gradients. The equations appear without any context, e.g., Q is not defined. They need to be introduced and explained much better.

3. All experiments are on 2 modalities, how does the method transfer to multiple modalities?

**Questions:**

1. Why does the experiments on MNIST in Table 3 achieve low accuracies ~85%? It is trivial to build a classifier that can achieve >95%.

2. What is the point of encoding audio using ResNets or even using ImageNet pretraining for audio? Is it not more realistic to use something more audio-specific? Related to this, what is the effect of using different architectures for different modalities?

3. Why can't we pre-train the uni-modal encoders separately and then perform the joint optimization as fine-tuning?

---

> ### Author Response · Authors · 2023-11-22
>
> Dear reviewer,
>
> We thank very much for taking the time to review our paper. We will improve this submission based on your valuable advice.
>
>
> Sincerely,
>
> Authors

---

### Official Review · Reviewer_ZL2h · 2023-10-31

**Soundness:** 3 good
**Presentation:** 3 good
**Contribution:** 2 fair
**Rating:** 5
**Confidence:** 4

**Summary:**

This paper concerns the multi-modal joint training task under the imbalance scenario, which leads the performance degeneration. To address this problem, they first identify that there are gradient conflict between multi-modal and uni-modal learning objectives, potentially misleading the optimization of shared uni-modal encoders. Considering that traditional Pareto method may fails in the context of multi-modal scenarios, they propose MMPareto algorithm, which could ensure a direction that is common to all learning objectives while preserving magnitude with guarantees for generalization. Experiments validate the effectiveness.

**Strengths:**

1.	The imbalance multi-modal learning is interesting and has real applications. The reviewer thinks that it is vital for multi-modal jointly training.
2.	The Pareto solution is solid for solving the imbalance in multi-modal jointly training and uni-modal training.
3.	The experiments on several tasks exhibit the superiority of the proposed method.

**Weaknesses:**

1. In Theorem 1, they claim that “If the sequence of training iteration generated by the proposed MMPareto method is infinite” However, the infinite for iteration is unrealistic, which is hard to satisfy in real scenarios. Can you provide a more solid Theorem?
2. The connections between the Theorem and experiments are hard to understand. More details are expected.
3. What are the assumptions behind the Theorem? More introductions are needed. For example, why the 2-order differentiable assumption can satisfy Lemma 2.
4. More details in experiments are needed. For example, the detailed introductions for "one joint loss" and "uniform baseline" should be added.
5. The reviewer is curious about the computational cost of finding the Pareto stationarity, more related experiments are expected.

**Questions:**

1. In Theorem 1, they claim that “If the sequence of training iteration generated by the proposed MMPareto method is infinite” However, the infinite for iteration is unrealistic, which is hard to satisfy in real scenarios. Can you provide a more solid Theorem?
2. The connections between the Theorem and experiments are hard to understand. More details are expected.
3. What are the assumptions behind the Theorem? More introductions are needed. For example, why the 2-order differentiable assumption can satisfy Lemma 2.
4. More details in experiments are needed. For example, the detailed introductions for "one joint loss" and "uniform baseline" should be added.
5. The reviewer is curious about the computational cost of finding the Pareto stationarity, more related experiments are expected.

---

> ### Author Response · Authors · 2023-11-22
>
> Dear reviewer,
>
> We thank you for your valuable review, which is essential for enhancing the paper's quality. We will next improve this submission based on your valuable advice.
>
>
> Sincerely,
>
> Authors

---

### Official Review · Reviewer_PLQt · 2023-11-06

**Soundness:** 3 good
**Presentation:** 3 good
**Contribution:** 3 good
**Rating:** 8
**Confidence:** 3

**Summary:**

This paper introduces the MMPareto algorithm, which addresses the challenge of imbalanced multi-modal learning. While previous methods have used targeted uni-modal constraints to improve performance, the authors identify a gradient conflict between multi-modal and uni-modal learning objectives that hinders optimization. This conflict is mitigated using the Pareto method, but traditional Pareto methods do not work well in multi-modal scenarios. The authors propose the MMPareto algorithm, which ensures a direction that benefits all objectives while preserving magnitude, thus improving model generalization. Empirical results across various datasets with different modalities demonstrate the superior performance of this method.

**Strengths:**

The paper demonstrates a high degree of originality. It identifies a previously overlooked challenge in multi-modal learning, the gradient conflict between multi-modal and uni-modal learning objectives.
The proposed MMPareto algorithm presents a novel approach to address this issue, offering a fresh perspective on multi-modal learning optimization.
The paper is well-written and effectively communicates its ideas. The introduction provides a clear problem statement, and the logical flow of the paper makes it easy for readers to follow. It provides theoretical analysis, clearly presenting the problem and the solution. Authors well motivated the challenges of traditional method and the necessity of bringing the MMPareto algorithm.
The algorithm is well-structured and theoretically grounded, and its implementation is evaluated across multiple datasets, showcasing its robustness.

**Weaknesses:**

The proposed method is established under the multi-task like multi-modal framework, especially with independent uni-modal encoders. It is a fundamental limitation of this discussion. The authors are encourged to provide discussion about the generalization of the idea to other models inplementing different fusion mechanism.

**Questions:**

Figure 4&5 is slightly hard to interpret due to the resolution. Authors can think of leaving the most important one but present it in the best quality while moving others to the appendix.

---

> ### Author Response · Authors · 2023-11-22
>
> Dear reviewer,
>
> We would like to express our sincere gratitude for the time and effort you have dedicated to reviewing our work.
> We will further refine this paper based on the valuable suggestions.
>
> Sincerely,
>
> Authors